# Multi-Scale Fully Convolutional Network-Based Semantic Segmentation for Mobile Robot Navigation

**Thai-Viet Dang** [1,*] and **Ngoc-Tam Bui** [2]

1   Mechatronics Department, School of Mechanical Engineering, Hanoi University of Science and Technology, Hanoi 10000, Vietnam
2   Shibaura Institute of Technology, Tokyo 135-8548, Japan
*   Correspondence: viet.dangthai@hust.edu.vn; Tel.: +84-0989458581

**Abstract:** In computer vision and mobile robotics, autonomous navigation is crucial. It enables the robot to navigate its environment, which consists primarily of obstacles and moving objects. Robot navigation employing impediment detections, such as walls and pillars, is not only essential but also challenging due to real-world complications. This study provides a real-time solution to the problem of obtaining hallway scenes from an exclusive image. The authors predict a dense scene using a multi-scale fully convolutional network (FCN). The output is an image with pixel-by-pixel predictions that can be used for various navigation strategies. In addition, a method for comparing the computational cost and precision of various FCN architectures using VGG-16 is introduced. The binary semantic segmentation and optimal obstacle avoidance navigation of autonomous mobile robots are two areas in which our method outperforms the methods of competing works. The authors successfully apply perspective correction to the segmented image in order to construct the frontal view of the general area, which identifies the available moving area. The optimal obstacle avoidance strategy is comprised primarily of collision-free path planning, reasonable processing time, and smooth steering with low steering angle changes.

**Keywords:** computer vision; fully convolutional networks; mobile robot; navigation; obstacle avoidance; semantic segmentation

## 1. Introduction

Mobile robots navigate safely through their environments by detecting obstacles and moving objects in real-time. Standard navigation assistance systems utilize sensors such as sonars [1], laser scanners [2], IR sensors [3], and cameras [4–6] to detect obstacles. Wall-following, edge detection, and line-following are examples of conventional navigation methods [1,2,7].

Cameras and Lidars are two of the most popular and effective solutions [6,8]. Lidars provide depth information in all directions, maintaining a perfect approximation of the world, despite being prohibitively expensive and requiring a relatively high level of computing power [6,9]. On the other hand, cameras provide low-cost scene information to detect any small footprint [2,4]. Due to the prevalence of monocular cameras with affordable prices and high precision, the proposed solution has eliminated previously existing drawbacks. Real-time image segmentation and autonomous navigation have been accomplished successfully.

Vision-based indoor mobile robot navigation has become a popular sensing method for autonomous navigation due to its ability to provide detailed information about the environment [3,10–13] that may not be obtainable using combinations of other types of sensors. Semantic segmentation by means of deep learning (DL) is a crucial task in computer vision, with numerous applications including scene understanding, robotic perception, and image compression [14–23]. For example, images captured by aerial

cameras can be utilized in urban planning. In the long-term vision, real-time data will be provided to design and develop a town or city. In addition, object detection and traffic flow analysis technologies facilitate urban traffic management. With the advancement of imaging technology, semantic segmentation will precisely define semantic classes, such as buildings, transportation infrastructure, trees, and low vegetation [16–23]. However, the quality of segmented images from aerial cameras requires both remote sensing and computer science expertise. Minae et al. [20] investigated the relationships, benefits, and challenges of these DL-based segmentation models. Li et al. [21] compared the basic dataset used in various structures to provide the essential methods of semantic segmentation by contrasting the datasets. To address the dearth of standardized datasets for evaluating object segmentation. In [22], the 2D semantic labeling contest is recommended as a solution to this problem. The output of segmentation could combine multiple binary masks to segment the input image into distinct classes. In addition, multi-class semantic segmentation had achieved remarkable results for mobile robot path planning in complex environments with numerous obstacles. Fusic et al. [23] introduced scene terrain classification using a vision sensor-based DL algorithm. The proposed classification algorithm distinguishes between terrain and obstacles by analyzing the acquired image datasets. However, mobile robot navigation utilizing scene terrain classification had not been demonstrated. To conserve memory resources and ensure processing speed, we utilized binary semantic segmentation for available and unavailable regions. The architecture of the segmentation model has become lighter and faster, with quality assured.

Recent advancements in deep learning have allowed us to unlock the segmentation potential of the renowned convolutional neural network VCG. Particularly, the authors design FCN-VGC-16 based on the VCG-16 model to solve the semantic segmentation task [24] and achieve efficient scene understanding for autonomous navigation. It includes an encoder and a decoder. The authors employ a previously trained model from prior networks for the encoder. The authors then use multi-scale fusion to generate a dense scene prediction in the decoder block. The method captures natural indoor settings effectively.

In addition, the optimized path-finding algorithm is implemented for a single camera system. Therefore, data processing requires sufficient performance and speed rate despite limited resources. Therefore, the authors design and build the binary semantic segmentation for two classes, such as available and unavailable regions. On the basis of the segmentation model's output, the authors can define the available area that will serve as input to the autonomous mobile robot navigation system. The authors achieve an overall accuracy of 93.5 percent with a new dataset of images from the Ta Quang Buu library. Finally, the authors successfully apply perspective correction to the segmented image in order to construct the frontal view of the general area, which detects the available area for movement in real-time. In order to solve complex problems in real-world environments, obtained results could also be updated and integrated with sensor systems [3].

The structure of the study contains the following sections. Recent works on semantic segmentation are presented in the section titled Related Studies. The Section titled Binary Semantic Segmentation FCN-VGG-16 explains our method's overall architecture. In conclusion, experimental results demonstrate the obtained results and compare our approach to previous approaches in the field of semantic segmentation.

## 2. Related Studies

Deep learning is superior to conventional semantic segmentation methods due to its key advantages. The semantic segmentation of images must comprehend their contents. Various semantic segmentation applications included autonomous vehicles, which required a precise understanding of their environment at the pixel level [16–25].

Recent achievements in deep learning, particularly convolutional neural networks (CNNs), have prompted a shift in computer vision research toward learning-based approaches [26]. The first convolutional neural network, LeNet, was applied to the recognition of ten handwritten digits [27]. AlextNet introduced Imagenet's large-scale visual recog-

nition in 2012 [28]. Typical CNNs such as VGG [29], Inception [30], ResNet [31], and DenseNet [32] have been proposed and constructed based on their success. The networks had exerted significant effort to enhance their optimization capability.

Deep convolutional neural networks (DCNNs) have become rapidly applicable to a variety of machine learning tasks [32–35]. They have self-learned representations using acquired data. Several deep architectures, namely fully convolutional networks [24], SegNet [32], and ENet [33], were recommended to perform pixel-wise segmentation and demonstrated outstanding performance. The FCN encoded the original image as a collection of feature maps containing only semantic information regarding objects and textures. For semantic segmentation, Badrinarayanan et al. [32] presented a SegNet to construct a semantic segmentation model under different environments. The FCN proposed by Sun et al. [34] on the ISPRS Vaihingen benchmark achieved an overall accuracy of 90.6%. Lin et al. [35] employed conditional random fields with CNN-based pairwise potential functions to identify semantic correlations in popular semantic segmentation datasets. Oršic et al. [19] successfully trained and operated in real-time on low-power embedded devices with very large semantic segmentation models. The authors constructed binary semantic segmentation based on multi-class semantic segmentation using FCN using indoor autonomous mobile robot navigation. In the section titled Related Studies, the authors examined each of these methodologies in turn.

### 2.1. Fully Convolutional Networks (FCNs)

Upon the completion of convolutional neural networks, a fully convolutional network was constructed. The fundamental idea utilized skip-layer fusion to decode decision characteristic maps for pixel-smart prediction. Consequently, the output results enabled the community to examine the entire process. Layers of transposed convolution with trainable pa-parameters were used to accomplish upsampling. To improve the accuracy of the output, the upsampled outputs of a specific layer were concatenated with the outputs of the previous layer. Consequently, spatial statistics from the more superficial layers were combined with deep functions from the deeper layers. Since FCNs lacked a fully connected layer, the network could accept arbitrary-sized input images, whereas fixed-sized inputs restrict CNNs. Moon et al. [28] utilized AlexNet architecture that achieved ILSVRC12, as well as VGG [29] and GoogLeNet [30,31], which achieved exceptionally positive results in ILSVRC14.

Each of these classification networks was not only used for classification tasks, but also as an image encoder. Long et al. [24] presented a new FCN architecture for semantic segmentation. Using any CNN architecture without fully-connected layers, Long et al.'s FCN encoded a raw image as the input data into a set of feature maps that retained only semantic information about objects and textures. In ISPRS 2D semantic labeling competitions, the leading teams employed numerous variants of fully conditional networks that established state-of-the-art Vaihingen datasets. Badrinarayanan et al. [33], Sun et al. [35], Lin et al. [36], and Sherrah et al. [37] successfully applied FCN schemes to image segmentation and obtained satisfactory outcomes.

### 2.2. SegNet

SegNet was utilized by Mong et al. [28] and Badrinarayanan et al. [33] for primary scene understanding. Therefore, it was designed to be a memory and machine time efficient during abstract thought. Since decoder layers only use scoop pooling indices from corresponding encoder layers to perform distributed upsampling, the network had fewer trainable parameters. SegNet adopted VGG-16 as a framework for encoder-decoders and eliminated the connected layers [29]. The decoder sub-network could be a thirteen-layer mirror image of the encoder sub-network. SegNet provided reasonable performance with competitive inference time and was superior to other architectures in terms of memory. SegNet was a credible candidate for real-time applications and was adopted on computers with limited computing power.

## 2.3. Efficient Neural Network (ENet)

In mobile phone applications, the ability to perform pixel-aware semantic segmentation in real-time is essential. ENet aims to reduce inference time for snapshots by reducing the number of floating-factor operations present in earlier architectures. This is also an encoder–decoder based primarily on architecture, with the decoder being significantly larger than the encoder. He et al. [32] adopted a concept from ResNet based on complete architectures. There was a major unmarried branch with the convolutional filter to separate and recombine with element-smart addition. The architecture no longer employed any biased language and it produced no inaccuracy. They also employed early downsampling, which decreased the image dimensions and thus saved on costly computations at the outset of the community. Using dilated convolutions also increased the community's receptive field size.

## 3. Binary Semantic Segmentation FCN-VGG-16

Binary semantic segmentation FCN-VGG-16 was mainly divided into two major parts: (1) FCN-VGG-16 architecture and (2) model training. Those two parts are described further in the following two sections (Sections 3.1 and 3.2).

### 3.1. Network Architecture

The authors implement a network primarily based on the concept of FCN models to perform immediate pixel-specific labeling. In addition, the authors have an excellent segmentation result. Similar to the FCN variants, the community's foundation is VGG. The authors chose VGG over AlexNet because it is a significantly different generalized model, allowing for more accurate predictions. In contrast to previous paintings utilized by Long et al. [24], the authors employ deconvolutional layers on a total of four scales. The authors discover that using different scales and substantial margins aid in refining their predictions.

In Figure 1, as proposed by Yang et al., the architecture initially forecasts a rough output at a low resolution, and then gradually refines it by fusing with preceding layers to provide both local and global reasoning. The color grey represents the input layer, which is the RGB image. Blue indicates convolution operations followed by nonlinearity; the authors employ a rectified linear unit in this instance (ReLU). The authors employ a series of convolution operations to extract additional features, as is the standard in VggNet. Using zero padding, the convolution operations do not alter spatial dimensions. Red indicates a maximum pooling operation. The max pool serves a simple purpose: it reduces the spatial dimension by selecting the most significant values using a $2 \times 2$ sliding window across the feature maps. During the network's forward pass, a total of five max-poolings are utilized.

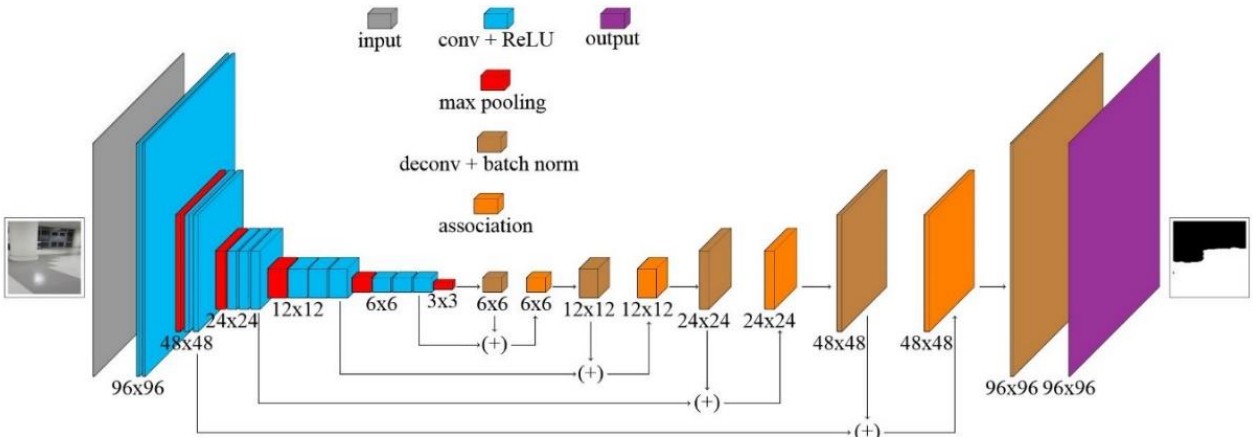

**Figure 1.** Proposed multi-scale fully convolutional network.

Figure 2 depicts VGG-16 as an encoder block made up of convolutional and max-pooling layers. The decoder is merely the reverse procedure with some modifications. Brown represents the transposed convolution operation followed by batch normalization. Upsampling low-resolution feature maps to a higher resolution is the primary function. Orange represents layer fusion, in which a high-level feature map is combined with a low-level feature map to provide both local and global reasoning. This procedure is repeated four times until the initial spatial dimension is restored.

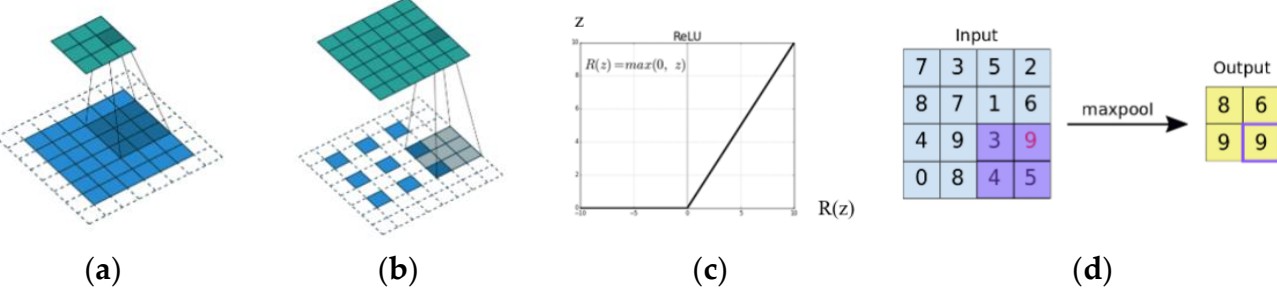

**Figure 2.** Proposed architecture with (**a**) convolution demo (F = 3, S = 2, P = 1); (**b**) deconvolution demo (F = 3, S = 2, P = 1); (**c**) ReLU (rectified linear unit); and (**d**) max pooling.

For activation, each of the hidden layers employs rectified linear units. The fully connected VGG-16 layers are eliminated from the network. The authors combine features extracted from multiple layers for the decoder block using multi-scale fusions. The input to the network is a $96 \times 96 \times 3$ RGB image, and the output of the first scale is 1/32 the size of the input. The authors perform transposed convolution (deconvolution) to a 1/16 scale and combine it with a conv13($6 \times 6 \times 512$) layer to obtain the prediction for the second scale. The authors repeatedly upsample and combine it with the conv10($12 \times 12 \times 256$) layer to obtain the third 1/8 scale output. As the authors continue upsampling the output of the corresponding convolutional layer, the network grows. Lastly, the upsampled step will achieve the desired image size of $96 \times 96 \times 64$, followed by a classifier that transforms the shape to $96 \times 96 \times C$, where C is the number of classes that the authors wish to semantically segment. Since the classifier is added to ensure that each pixel belongs to the ground or non-ground classes, the authors chose C = 2 in this instance. For the majority of robot navigation purposes, ceilings, doors, and pillars were not deemed unimportant.

*3.2. Model Training*

Our experiments are conducted on a server with the following specifications using Python 3.11.0 with the Tensorflow 1.4 framework: a computer with an Intel(R) Core(TM) I7-8750h CPU @2.20ghz 2.21ghz, 8.00GB of RAM, a 64-bit operating system, and Windows 10 Home English. The authors employ the most prevalent metric for semantic segmentation: mean intersection over union (mIoU). Training is performed by optimizing the binary cross-entropy loss (BCE):

$$\text{BCE} = - \sum_{i=1}^{C=2} y_i \log(\hat{y}_i) = -y_1 \log(\hat{y}_1) - (1 - y_1) \log(1 - \hat{y}_1) \tag{1}$$

where $\hat{y}_i$ is the class SoftMax probability produced by our network and $y_i$ is the ground truth of the corresponding prediction.

The focus of the paper is the corridor environment, which mobile robots operating indoors must traverse frequently. The authors actively collect training datasets specifically for corridors for this reason. Yang et al. [38] collected the first dataset containing 967 images from three sources. The first source is SUN RGBD [39] (category "corridor") with 349 images, the second source is SUN database [40] (category "corridor") with 327 images, and the third source is self-collected videos from the Carnegie Mellon University campus with 291 images. The second dataset is from Tsai et al. [41] and consists of four video sequences

captured with a camera mounted on a wheeled vehicle; however, we must relabel them to suit our needs. Numerous images within these two datasets are straightforward and straightforward to segment. It is more difficult for a mobile robot to navigate in an environment with numerous obstacles and occasionally obscured backgrounds. Thus, the authors self-collected several images that they deemed somewhat challenging to reflect on. One hundred images were taken from the Ta Quang Buu Library, and two hundred more were selected with care from the Internet. The authors made use of available annotations and annotated manually using LabelMe [42]. The final dataset contains 1470 fully annotated images. The dataset is divided into 1176 training images (80%) and 294 testing images (20%) for benchmarking purposes. All images are resized to 96 by 96 pixels and annotated with polygons that correspond to two classes: ground and non-ground.

To optimize the binary cross-entropy defined by Equation (1), network parameters are learned using one of the stochastic gradient descent (SGD) variations. In contrast to Yang et al.'s proposal [38], our network's capabilities were expanded through the use of data augmentation, where images for training are in their original forms. Figure 3 depicts how the authors apply random horizontal flipping and color shifting to training images before passing them through our network.

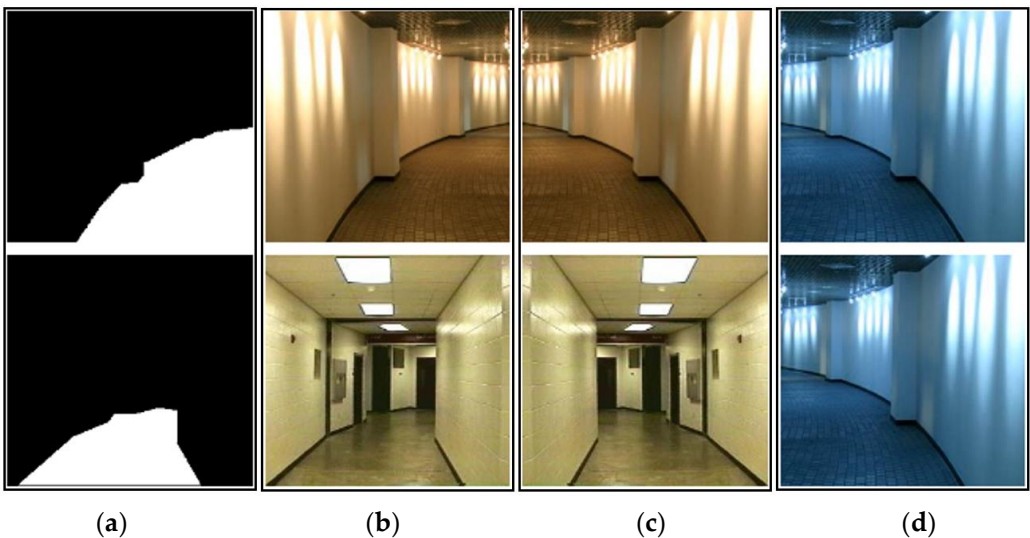

|  (a)  |  (b)  |  (c)  |  (d)  |

**Figure 3.** Data augmentation for training images. (**a**) Ground truths; (**b**) raw images; (**c**) horizontal flipping images; and (**d**) color-shifting images.

Transfer learning is utilized extensively to expedite training. The authors initialize the network parameters utilizing pre-trained VGG models; the only challenging aspect is the selection of training settings. As depicted in Figure 4, the authors select the following configuration after training and validating multiple network variants. The batch size chosen by the authors is sixteen, with a learning rate of 0.001 and a momentum of 0.9. Every 80 epochs, the learning rate diminishes by 0.1. Training and validation are performed simultaneously. Each training process is optimized for 300 epochs of convergence.

To accelerate the training process, Figure 4a depicts the validation mIoU as the noise affecting the training process's quality. Consequently, the obtained prediction will diminish. Therefore, the authors utilize data augmentation to prevent overfitting. The technique facilitates the refinement of Figure 4b's prediction. In addition, the quality of the prediction will be greatly enhanced with virtually no noise. Probabilistic models in [43] complete the segmentation noise filtering process.

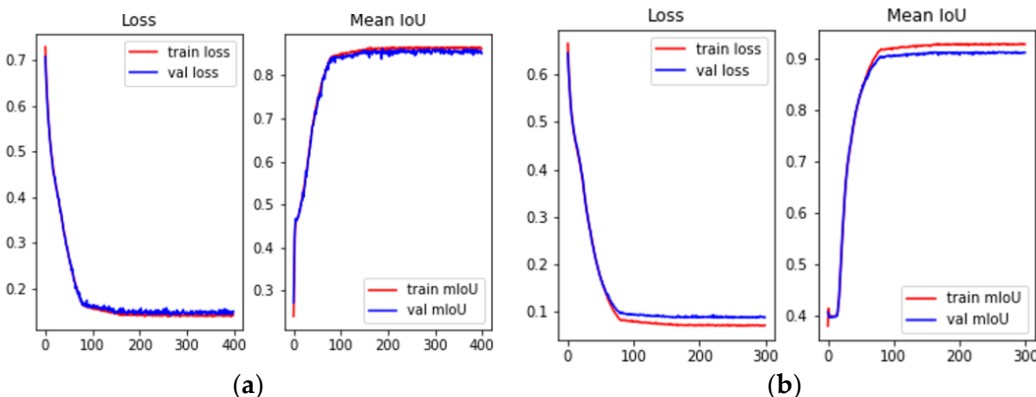

**Figure 4.** Training and validation on training images using mean intersection over union, mIoU, as metric. (**a**) VGG-FCNs-CMU-96 × 96 and (**b**) VGG-FCNs-Augment-96 × 96.

## 4. Experimental Results

The authors evaluate the proposed method by applying it to 294 test images. In addition, the authors collect additional real images to enhance the dataset. They include both previously collected, self-collected, and annotated images, for a total of 1470 images. The authors make use of mIoU for the semantic segmentation task and a comparison with prior methods. The authors display qualitative and quantitative results alongside analysis. In conclusion, the results of constructing a frontal view of the floor plane demonstrate the importance of semantic segmentation in the navigation and obstacle avoidance strategy of mobile robots.

### 4.1. Qualitative Results

As previously stated, mobile robots are typically unable to identify the wall plane boundary under conditions including poor lighting, viewpoint variation, background clutter, and illumination. This section illustrates several of these scenarios to evaluate the performance of our proposed network. Figure 5 depicts four schemes in which the optimal condition is addressed first, followed by more difficult circumstances. Our proposed network is able to correctly classify ground and non-ground regions in low-light conditions, a situation that typically causes difficulty for humans. This can be explained by the fact that the authors undergo color-shifting training, which enables our network to function effectively in the dark. Since corners and intersections are ubiquitous within the indoor environment, the following example is more obvious to mobile robots. The authors conduct the final network performance test on an image containing numerous objects in the background. Our network accurately predicts the ground boundary in these difficult circumstances, demonstrating its robustness across corridor types. Figure 6 depicts the experimental results of segmentation in the environment of the Ta Quang Buu library.

### 4.2. Quantitative Results

By propagating through several alternate convolutional and pooling layers, the resolution of the output feature maps is reduced in this particular FCN. Consequently, direct FCN predictions are typical of low resolution. We first compare our semantic segmentation results to those of prior researchers in Table 1. Our experiments are conducted using the same dataset of binary class images as previous models. Since Hoiem et al. [44] also combine geometry modeling and learning, we retrained the surface classifier using our new training data. Comparing our method and [44] to the same viewpoint, it is more robust than in an indoor environment [45,46], which may not detect valid vanishing points or sufficient line segments to form workable room models. In addition, authors compare FCN with ResNet since the encoder is typically a classification network such as VGG, GoogleNet, or ResNet. FCN using ResNet 101-4s and ResNet101-8s is superior to the model by Yang et al. As shown in Table 1, our best model outperforms Yang et al. [38] by 6% and is 5% better than

FCN-ResNet 101 regarding mIoU. Our network could learn a more accurate representation of the environment and thus improve its performance as a result of the expansion of our dataset with more difficult images and the extensive use of data augmentation in training.

When moving into the interference area of obstacles, the frontal view may be restricted if the indoor environment contains a large number of obstacles. Consequently, the performance and speed rate of the binary semantic segmentation model satisfies the requirements of [19], Yang et al. [38], and our obtained result using VGG-16 in Table 2.

Table 3 presents the quantitative outcomes of our network in response to variations in the encoder block. Here, the authors compare four VGG-16-based architectures: FCN32s, FCN16s, FCN8s, and FCNs. The former predicts using deeper layers, whereas the latter employs both shallower and deeper layers.

All variants employ the VCG-16 encoder block. Table 3 demonstrates that using different scales significantly improves prediction accuracy. Moreover, with our FCN-VGG-16, the accuracy rate increases from 91.8% to 93.5%. The authors can conclude that using a multi-scale fully convolutional network improves the accuracy of segmentation because deeper layers capture local information and shallower layers capture global information. Additionally, the noise introduced by segmentation will hinder the ability to avoid obstacles. In addition, the conditional random fields probabilistic model is used to filter and eliminate output segmentation noise [43]. The output of binary semantic segmentation FCN-VGG-16 ensures optimal path planning for autonomous mobile robots.

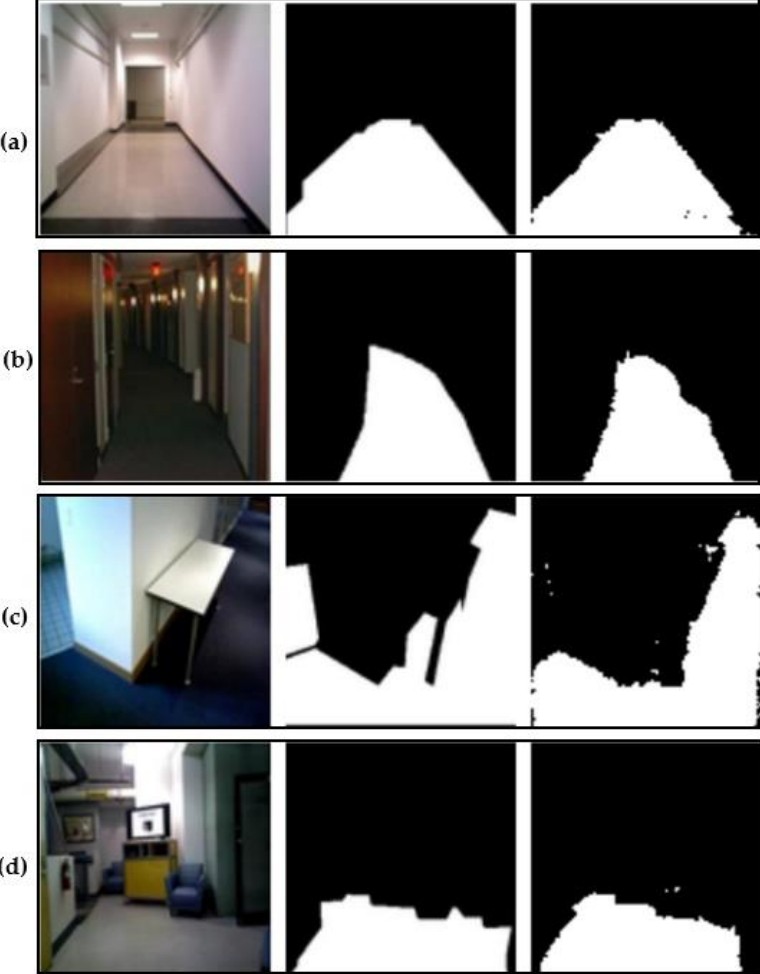

**Figure 5.** Segmentation results with raw images, ground truths, and predictions under various conditions. (**a**) The best condition with no obstacles; (**b**) poor lighting condition; (**c**) intersection; and (**d**) clutter background.

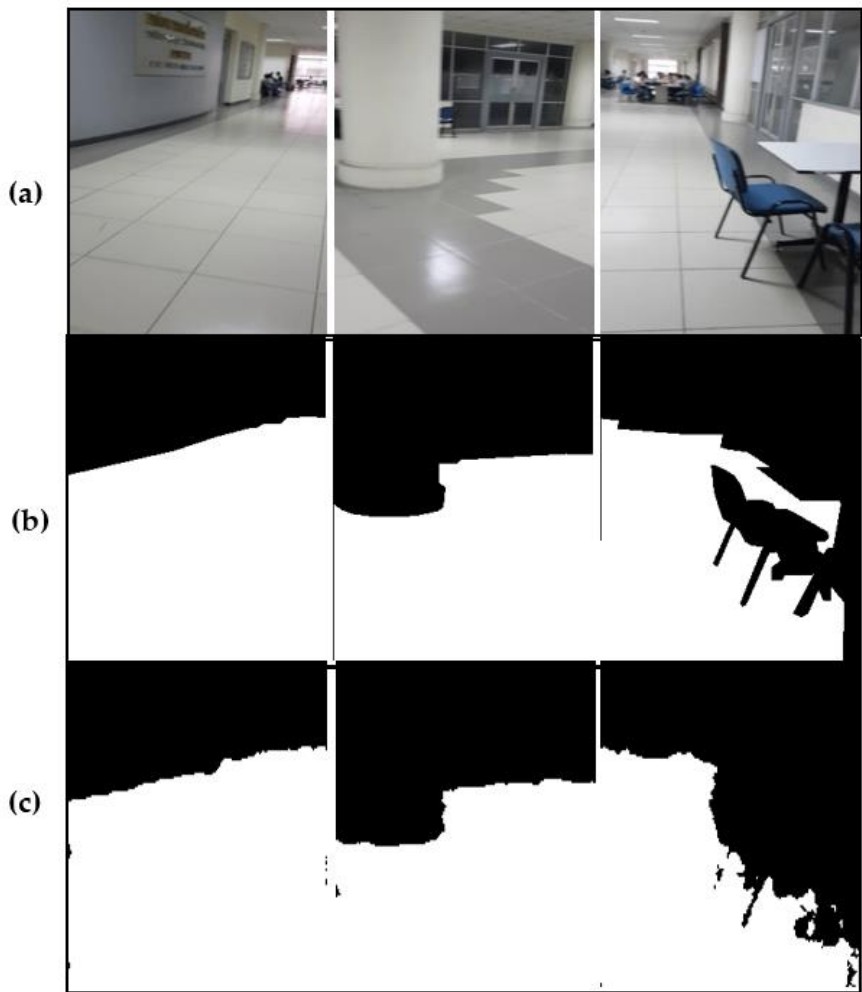

**Figure 6.** Segmentation results in the real environment (Ta Quang Buu library). (**a**) Raw images; (**b**) ground truths; and (**c**) predictions.

**Table 1.** Comparing our FCN to previous works using the same dataset.

| Methods | CNN Architectures | mIoU |
|---|---|---|
| Lee [41] | Alexnet | 68.38% |
| Hedau [42] | Alexnet | 69.6% |
| Hoiem [40] | Alexnet | 70.12% |
| FCN [38] | Alexnet | 87.10% |
| FCN [24] | ResNet 101-8s | 88.1% |
| FCN [24] | ResNet 101-4s | 88.3% |
| **Our FCN-VGG-16** | **VGG-16** | **93.5%** |

**Table 2.** Comparing the outcomes of various models to mIoU and fps.

| Model | mIoU | fps |
|---|---|---|
| ResNet 18 [33] | 88 | 35 |
| Mobile net V2 [33] | 88.5 | 36 |
| Yang et al. [38] | 87.1 | 30 |
| **Our FCN-VGG-16** | **93.5%** | **37** |

**Table 3.** Comparative analysis of FCN variants.

| Variants | mIoU |
|---|---|
| FCN32s | 91.88% |
| FCN16s | 92.93% |
| FCN8s | 93.43% |
| **Our FCN-VGG-16** | **93.5%** |

*4.3. Practical Results*

Using the obtained results based on the fully trained model of binary semantic segmentation FCN-VGG-16, the authors design the following navigation and obstacle avoidance strategy (see Figure 7). Step 1: Sematic segmentation (detect available path) in Figure 7a.

Step 2: Frontal view of floor plane (map of the area) in Figure 7b.
Step 3: Detecting the collision-free area in Figure 7c.
Step 4: Constructing the shortest path in Figure 7d.
Step 5: Ensuring the collision cost in Figure 7e.
Step 6: Ensuring the smooth cost to create the smoothed path in Figure 7f.

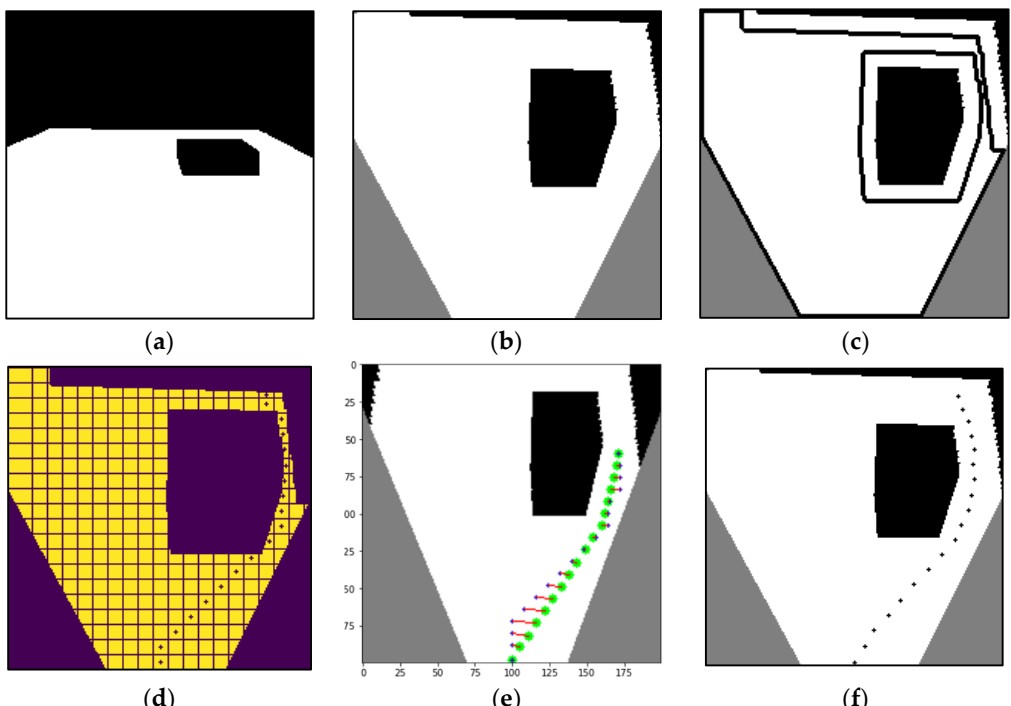

**Figure 7.** Obstacle avoidance navigation strategy. (**a**) Sematic segmentation; (**b**) frontal view; (**c**) collision-free area; (**d**) finding shortest path based on A* algorithm; (**e**) using collision cost; and (**f**) using collision cost and smooth cost.

After performing binary semantic segmentation, the authors obtained the images in Figure 8 labeled with the available area for movement (white) and unavailable space (black). The perspective of captured images produces significant distortion. So, measuring the distance to the obstruction was quite difficult. In the frontal view of the floor plane, the image of a checkerboard is utilized to approximate the transformation matrix.

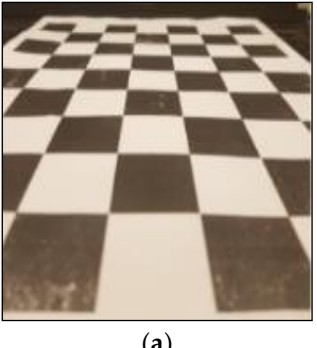
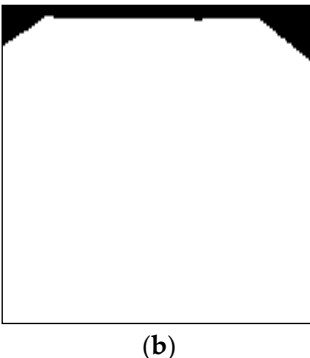

(**a**)  (**b**)

**Figure 8.** Results of semantic segmentation. (**a**) the frontal view using the image of checkerboard; (**b**) the obtained frontal view of the floor plane.

In order to plan the trajectory, the authors require proportional ground-based coordination. Based on the mathematical representation of the camera and the pinhole model, the authors can approximate the relationship between world and image coordinates using simple matrix multiplication. For p and P to be the image point and the world point, respectively, the following describes the general transformation from the world point to the image point:

$$p = M_{int} \times M_{ext} \times P \tag{2}$$

where $M_{int}$ is the matrix of intrinsic parameters and $M_{ext}$ is the matrix of extrinsic parameters. The intrinsic parameters are internal and fixed to a particular camera setup, while the extrinsic parameters are external to the camera and may change with respect to the world frame.

The intrinsic and rotational components of extrinsic parameters will remain constant due to the camera setting, which is attached to the mobile device with a fixed angle of view and will not change during movement. Consequently, the approximation need only be computed once.

We proposed that our image only describes the ground plane with the available and unavailable area; the authors can construct the frontal view of the ground plane using the transformation matrix and four points from [47]. The following defines the projective transformation based on four points.

Let $(x,y)$ and $(x',y')$ coordinates of a pair of matching points $x$ and $x'$ in the world and image plane. The authors give n-point correspondences satisfying $(x,y) \leftrightarrow (x',y')$. Then, compute the transformation $H$ such that:

$$x' = Hx_i. \tag{3}$$

because each point correspondence gives two constraints:

$$x' = \frac{x_1'}{x_3'} = \frac{h_{11}x + h_{12}y + h_{13}}{h_{31}x + h_{32}y + h_{33}}; y = \frac{x_2'}{x_3'} = \frac{h_{21}x + h_{22}y + h_{23}}{h_{31}x + h_{32}y + h_{33}} \tag{4}$$

Therefore,

$$x'(h_{31}x + h_{32}y + h_{33}) = h_{11}x + h_{12}y + h_{13}, \text{ and } y'(h_{31}x + h_{32}y + h_{33}) = h_{21}x + h_{22}y + h_{23}. \tag{5}$$

Finally, if $n \geq 4$ (no points of n are collinear), $H$ is determined uniquely. Firstly, the authors give the coordinates of four points on the scene plane in Figure 9. Secondly, the converse of this is that it is possible to transform any four points in a general position to any other four points in a general position by a projective transformation in Figure 9.

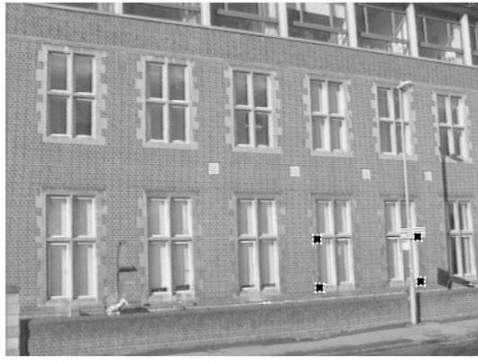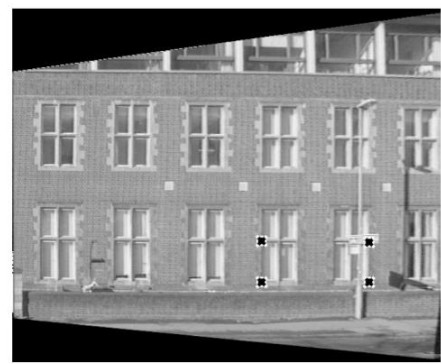

**Figure 9.** Four points on the scene plane define a projective transformation.

Hence, the authors perform the approximation based on correspondence points in arbitrary and frontal views of the same checkerboard in Figure 10.

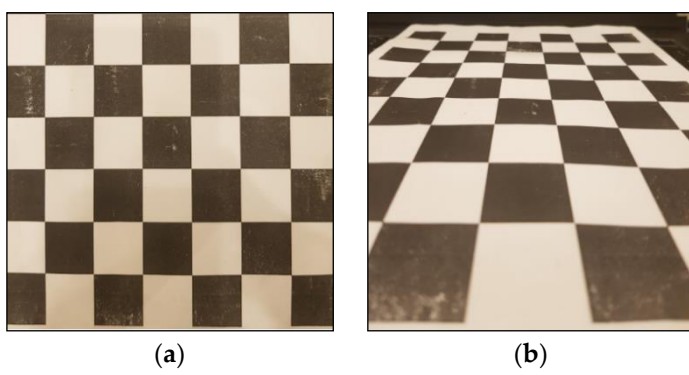

(**a**)                    (**b**)

**Figure 10.** Reference plane and source image.

Based on the aforementioned transformation in Equation (5), the authors construct the frontal view of the floor from an arbitrary view of the available area proportional to the ground coordinates in Figure 11. The transformation will result in a plane whose dimensions are proportional to global coordination. The image's results can be used to determine the object's distance.

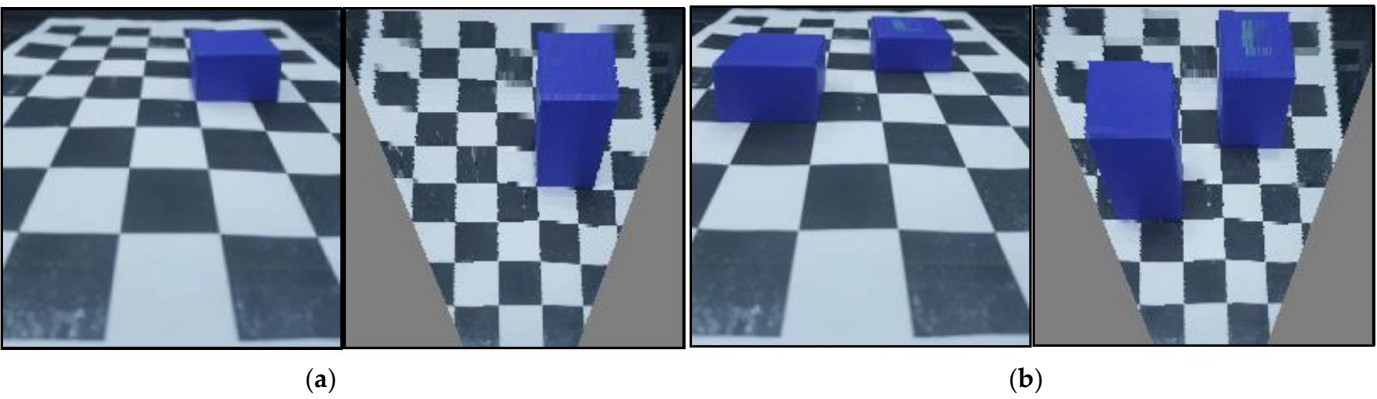

(**a**)                                        (**b**)

**Figure 11.** Frontal view of the plane with (**a**) a single object and (**b**) multiple objects.

The experimental environment used an Intel(R) Core(TM) I7-8750h CPU @2.20ghz 2.21ghz, 8.00GB of RAM, a 64-bit operating system, Windows 10 Home English, and Python 3.11.0. The autonomous mobile robot is equipped with a Jetson nano 4 GB and Raspberry Pi Camera Module V2 8MP using Sony IMX219 8 Megapixel, as seen in Figure 12.

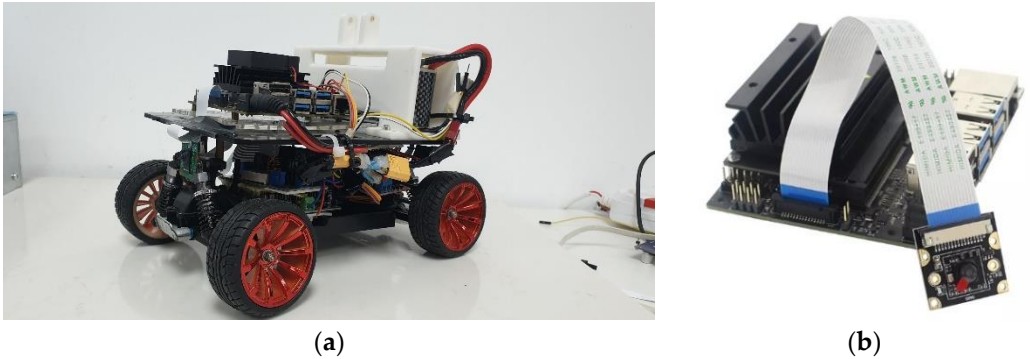

(**a**)                                                     (**b**)

**Figure 12.** A mobile robot equipped with a Jetson nano and Raspberry Pi Camera Module V2 8MP.

The authors evaluate the transformation results and confirm the crucial role of semantic segmentation in constructing the floor's frontal view. Then, the optimal path planning for the mobile robot will be successfully designed. The actual experimental outcomes strengthen the process of collision-free zone detection. When the frontal view is standardized, a comprehensive navigation and obstacle avoidance strategy will be formulated. To test the efficacy of our proposed sematic segmentation with four-wheel mobile robot navigation, the following experiments were conducted in Figure 13's 2.8 m × 1.4 m environment containing four obstacles. Figure 13 demonstrates that our optimal obstacle avoidance strategy [48] plans a collision-free global path for the mobile robot, which is then followed by the mobile robot as it moves.

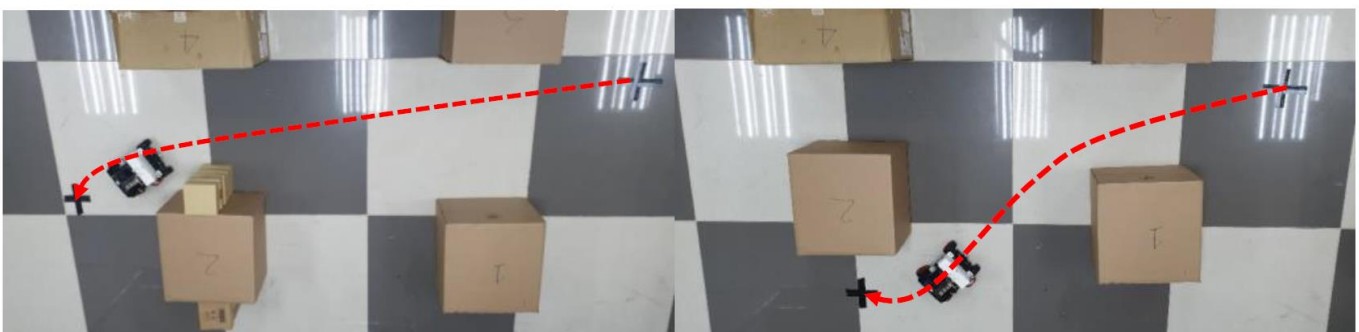

**Figure 13.** The mobile robot's path in four obstacle scenarios.

Because inaccurate segmentation affects obstacle avoidance results, in order to build a probabilistic model to segment and label sequence data, the results of the segmentation model must be pre-processed with conditional random fields. Segmentation's noise is completely filtered [43]. To increase the safety of the A* path-finding in Figure 14, the given environment modeling will be revised to keep the path clear of obstacles (blue line). To address this issue, the authors will add an additional collision cost to the evaluation function of A*. If the conventional A* algorithm is utilized, the mobile robot may collide (black line) when passing an obstacle or turning around it.

After independently examining each component, the authors combine the cost function and produce the final results. The mobile robot avoids obstacles safely and efficiently in order to reach its destination. The authors evaluate the performance and quality of trajectories by measuring the processing time in Figure 15. The experimental simulation included the following two scenarios: Scenario 1 has one obstacle in the environment's center, while Scenario 2 has two obstacles in the environment's upper right corner. Using our optimal obstacle avoidance strategy described in [48], the processing time in scenario 1 is 0.0493 s and scenario 2 is 0.0725 s.

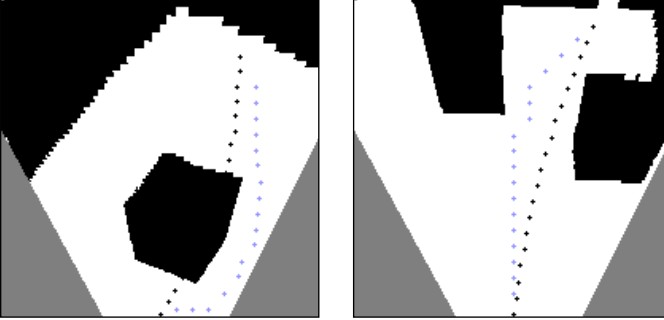

**Figure 14.** The path planning with collision, path, and smooth cost.

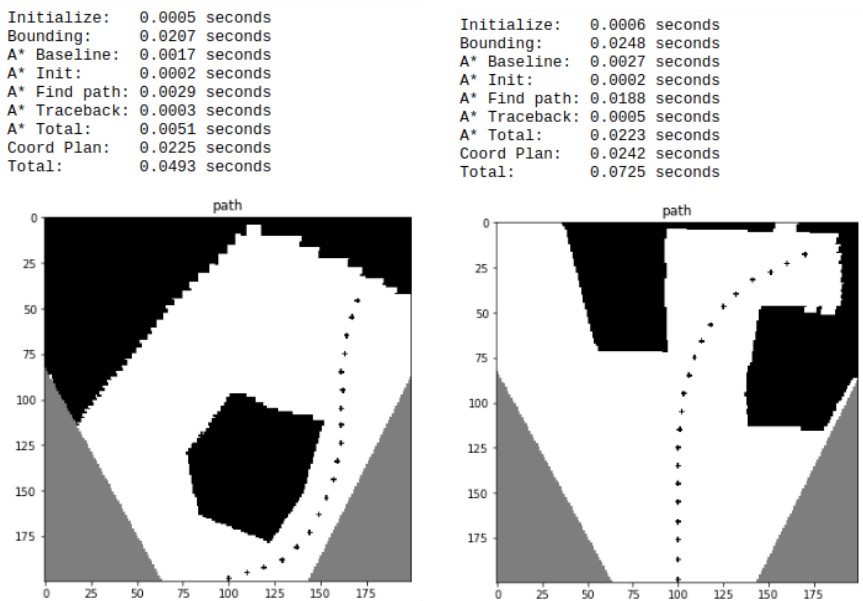

**Figure 15.** Final trajectory result with time measurement.

Furthermore, our obstacle avoidance strategy [45] ensured that the steering angle changes were less than 0.2 rad, as seen in Figure 16.

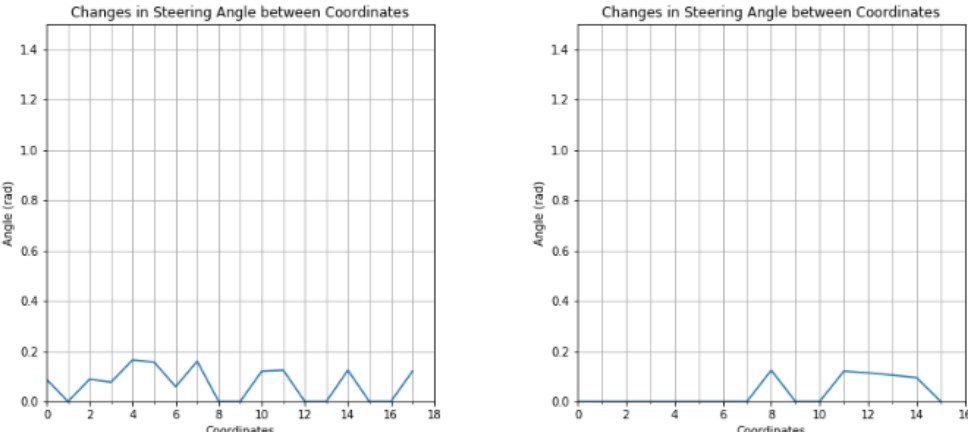

**Figure 16.** Changes in steering angle between coordinates.

Otherwise, the rotation angle on transformation changes would have a disproportionate effect on the precision of binary sematic segmentation, as seen in Figure 17.

| Angle (rad) | Accuracy |
|:---:|:---:|
| 0.00 | 100% |
| 0.01 | 95.93% |
| 0.03 | 89.92% |
| 0.05 | 85.31% |
| 0.07 | 82.31% |
| 0.09 | 81.05% |

| Angle (rad) | Accuracy |
|:---:|:---:|
| 1.57 | 98.97% |
| 1.51 | 89.42% |
| 1.47 | 80.96% |
| 1.42 | 73.48% |
| 1.36 | 65.56% |
| 1.31 | 51.55% |

(**a**)                    (**b**)

**Figure 17.** The influence of rotation angle on transformation (**a**) frontal place when rotating around the x-axis and (**b**) y-axis.

Consequently, binary sematic segmentation played an essential role in the navigation of mobile robots. In addition, the smoothed obstacle avoidance strategy maintained the quality of path planning and sematic segmentation results. When the standardization of the frontal view is completed, the entire navigation and obstacle avoidance strategy will be designed.

## 5. Conclusions

In this paper, the authors present a multi-layer, fully integrated network for binary semantic segmentation applied to autonomous mobile robot navigation. The principal component of our method is the use of multiple-layer information to efficiently classify ground and non-ground regions. The authors also demonstrated that data augmentation facilitates the prediction of challenging conditions, such as poor lighting and background clutter. The authors compiled a new dataset comprised previously collected and self-collected and annotated images for a total of 1470 images. The dataset can be used as a standard for corridor image segmentation in order to evaluate cutting-edge algorithms. Our method employs FCN architecture with VGG-16. With an mIoU of 93.5% and fps of 37, the proposed method outperforms other systems by a significant margin while maintaining a compact architecture. Based on the results of binary semantic segmentation, the optimal obstacle avoidance strategy consists primarily of collision-free path planning with reasonable processing time and low steering angle changes. In the frontal view, the navigation strategy for autonomous robots will be proactive in selecting flexible options, as opposed to relying on the choice of following an initial reference, as was the case with existing methods.

Future authors will enhance the multi-scale fusion technique for modern deep learning architectures, such as ResNet and MobileNet. Using depth-separated convolutional networks to reduce the model size and computational volume increases accuracy and processing speed. In the future, we would also like to use multi-class data to enhance binary semantic segmentation's ability to identify a variety of indoor obstacles. The authors would also like to improve the modeling of cluttered objects and wall planes to produce a more accurate and thorough scene interpretation. Consequently, the effectiveness of path planning will be enhanced in a variety of indoor environments. In order to solve complex problems in real-world environments, the obtained results could also be updated and integrated with sensor systems.

**Author Contributions:** Conceptualization, T.-V.D. and N.-T.B.; Project Adminitration, T.-V.D.; Supervision, N.-T.B.; Funding Acquisition: N.-T.B.; Methology, T.-V.D. and N.-T.B.; Formal Analysis, N.-T.B.; Software, T.-V.D.; Experimental data and system, T.-V.D.; Writing-Review and Editing, T.-V.D. and N.-T.B.; All authors have read and agree to the published version of manuscript.

**Funding:** This research is funded by Hanoi University of Science and Technology (HUST) under project number T2022-PC-029.

**Acknowledgments:** This research is funded by Hanoi University of Science and Technology (HUST) under project number T2022-PC-029. The School of Mechanical Engineering at HUST is gratefully acknowledged for providing funding, guidance, and expertise. This work was also supported by the Centennial Shibaura Institute of Technology Action for the 100th anniversary of Shibaura Institute of Technology to enter the top ten Asian Institutes of Technology.

**Conflicts of Interest:** The authors declare no conflict of interest.

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
