# Peer review of "Multi-Scale Fully Convolutional Network-Based Semantic Segmentation for Mobile Robot Navigation"

_electronics, doi:10.3390/electronics12030533_

Round 1

Reviewer 1 Report

- Sections 2 and 3 could be shorten.

- You should increase the number of test images to 10000. 

- Why are the images resized to only 96x96? Is it enough for your application? 

- Comment: https://doi.org/10.3390/robotics11060130

- Figures 5,6: There are some white spots in Figs. 5,6. Have you tried to remove them by applying post-filtering, e.g. morphological filters? How would such additional component affect your framework?

- You could add mean average precision.

Author Response

Response letter for resubmission of the paper to the Electronics (MDPI)

Section: Computer Science & Engineering

Special Issue: Machine Learning Methods in Software Engineering

Ph.D.Thai-Viet Dang

School of Mechanical Engineering.

Hanoi University of Science and Technology, Hanoi, Vietnam.

[01/04/2023]

Dear Editor of the Electronics

Special Issue: Machine Learning Methods in Software Engineering

Thank you very much for allowing us to revise the manuscript “Multiscale Fully Convolutional Network-based Semantic Segmentation for Mobile Robot Navigation.”  We want to thank the reviewers for their positive feedback and constructive comments. The reviewers’ suggestions and comments are beneficial to guide us in improving our manuscript.

We have addressed the reviewer’s concerns and believe the manuscript is significantly improved after making the suggested revision. Our detailed responses to the reviewer’s comments are added at the end of the revision. Changes in the revised manuscript are highlighted in color, corresponding to two reviewers’ comments. The revision has been carefully accomplished in consultation with all co-authors, and each author has approved the final version of this revision. Please see the attachment.

Thank you again for the consideration of our revised manuscript. We hope that the revised manuscript fulfills the requirements of the reviewers.

Best regards,

Thai-Viet Dang (On behalf of all authors)

RESPONSE TO REVIEWER 1 COMMENTS

Comment 1. Sections 2 and 3 could be shorten

Response 1: Dear reviewer, we checked carefully and the structure of each section in the paper is systematic.

Comment 2. You should increase the number of test images to 10000

Response 2: The main contribution of our paper is to design multi-scale fully convolutional network-based semantic segmentation for mobile robot navigation using low resource system. The reference [48] proved the feasibility of proposed method for mobile robot using monocular camera in terms of computational cost and accuracy. Other contribution is robust mobile robot navigation with small steering angle in tracking path planning. Hence, we self-collected images from inside Ta Quang Buu Library. With enough number of obtained images, semantic segmentation’s results will be will be the basis for successful construction of mobile robot navigation strategy in [48].

Comment 3. Why are the images resized to only 96x96? Is it enough for your application? 

Response 3: The main contribution of our paper is to design multi-scale fully convolutional network-based semantic segmentation for mobile robot navigation using low resource system. So, authors used the size of image 96x96. Is it enough and ensures the performance of mobile robot navigation strategy in [48].

Comment 4. Comment: https://doi.org/10.3390/robotics11060130. The paper tittles “Obstacles Avoidance for Mobile Robot Using Type-2 Fuzzy Logic Controller”.

Response 4: In the paper, Mohammad et al. introduced about the Takagi-Sugeno-Kang (TSK) algorithm to deal with different kinds of uncertainties in order to perform their tasks, such as tracking predefined paths and avoiding static and dynamic obstacles until reaching their destination. The main contribution is a design of a Type-2 fuzzy logic controller for Robotino movement. The modelling of Robotino is designed by Matlab 2021b. There IR sensors used to avoid collision. We will show some different contribution between our paper and Mohammad et al.’s paper as follows:

  • The main contribution of our paper is to design multi-scale fully convolutional network-based semantic segmentation for mobile robot navigation using low resource system. So we need only one monocular camera to collect image input. Then, detect the correct distance to obstacle for collision avoidance. In Mohammad et al.’s paper, authors can not show the solving simultaneously of three IR sensor data in the real time. Furthermore, in optimal mobile robot planning, the computational cost and performance plays an important role in low resource navigation system.
  • As for avoiding the dynamic obstacle, the experimental results had not enough evidence to prove the feasibility of Mohammad et al.’s method. The required data for the algorithm from three sensors are analog voltages which reflect the distance of the obstacle in cm. If existing disturbance, noise and uncertainty in the distance measuring values, authors should design TS fuzzy observer or integrate with the filter to ensure the fuzzy model’s input. A Type-2 fuzzy logic controller will handle the problem of the correction of distance in moving environment. Meanwhile, my paper focused on the optimal navigation under low resource system with suitable computational speed and performance of steering angle in tracking trajectory. The feasibility of our proposed method is enhanced in [48].
  • Because of the research field of Mohammad et al.’s paper is Robotics, practical results have been missing in the comparison with recent state of art methods.
  • Finally, the approximation in the TSK model should be considered in Mohammad et al.’s method. Three IR sensors are analog voltages will affect the quality of the Type-2 fuzzy logic. Mohammad et al. should pay attention to about the simultaneous processing of three IR sensors’ result to build Fuzzy rules. Finally, the obtained result of Mohammad et al.’s model will ensure the application of mobile robot navigation in different environments.

Comment 5. Figures 5,6: There are some white spots in Figs. 5,6. Have you tried to remove them by applying post-filtering, e.g. morphological filters? How would such additional component affect your framework?

Response 5: The main contribution of our paper is to design multi-scale fully convolutional network-based semantic segmentation for mobile robot navigation using low resource system. So the result of obtained image has been not absolutely perfect. We used Probabilistic Models in [43] to filter entirely the segmentation noise.

Comment 6. You could add mean average precision.

Response 6: The main contribution of our paper is to design multi-scale fully convolutional network-based semantic segmentation for mobile robot navigation. We used more comparisons with previous methods to prove the improvement of our method with suitable evaluation criteria in the experimental results.

Reviewer 2 Report

See the attached file for detailed comments.

Author Response

Response letter for resubmission of the paper to the Journal Electronics (MDPI)

Section: Computer Science & Engineering

Special Issue: Machine Learning Methods in Software Engineering

Ph.D.Thai-Viet Dang

School of Mechanical Engineering.

Hanoi University of Science and Technology, Hanoi, Vietnam.

[01/04/2023]

Dear Editor Journal of the Electronics

Special Issue: Machine Learning Methods in Software Engineering

Thank you very much for allowing us to revise the manuscript "Multi-scale Fully Convolutional Network-based Semantic Segmentation for Mobile Robot Navigation." We sincerely thank the reviewers for their positive feedback and constructive comments. The reviewers' suggestions and comments are beneficial to guide us in improving our manuscript.

We have addressed the reviewer's concerns and believe the manuscript is significantly improved after making the suggested revision. Our detailed responses to the reviewer's comments are added at the end of the modification. Changes in the revised manuscript are highlighted in color, corresponding to two reviewers' comments. The revision has been carefully accomplished in consultation with all co-authors, and each author has approved the final version of this revision. Please see the attachment.

Thank you again for the consideration of our revised manuscript. We hope that the revised manuscript fulfills the requirements of the reviewers.

Best regards,

Thai-Viet Dang (On behalf of all authors)

RESPONSE TO REVIEWER 2 COMMENTS

In this article, the authors presented a multi-scale fully Convolutional Network-based Semantic Segmentation for Mobile Robot Navigation. The author's selected topic is timely and interesting, but the current work possesses several drawbacks that must be considered to make this work capable of consideration in any journal. Some of the suggestions are listed below but are not limited to the mentioned ones.

Comment 1. A bulk of citations are added without any proper explanations, such as [3,10-13], [14-23], and so on. This must be avoided. Add only relevant references with proper explanations.

Response 1: In the Introduction, the references [3, 10-13] introduced the advantages of computer vision compared to other environment perception systems. Furthermore, Semantic segmentation using deep learning (DL) is crucial in computer vision. Hence, the authors used the references [14-23] to demonstrate the necessity of semantic segmentation applications in robotic perception and computer vision. The authors checked and proved the role of all the references in the manuscript.

Comment 2. The paper organization is written very badly. Please have a look at some standard articles and learn how to write article organizations.

 Response 2: Dear reviewer, thank you for carefully reading our manuscript and giving many constructive comments. Regarding the language of the paper, we plan to send our manuscript to the English native checker before submitting the final version for publishing. Moreover, all other requirements are revised based on the reviewer's comments; for detail, please find the highlighted revised colored text in the revision.

Comment 3. Figure 1 is completely invisible. Look like a snapshot. Similarly, in figure 2, and so on. Please add original figures.

Response 3: The main contribution of our paper is to design multi-scale fully convolutional network-based semantic segmentation for mobile robot navigation using a low-resource system. The reference [48] proved the feasibility of the proposed method for mobile robots using the monocular camera in terms of computational cost and accuracy. Another contribution is robust mobile robot navigation with a slight steering angle in tracking path planning. Hence, all results are original.

Comment 4. Please avoid the word "we" in the paper. For example, we did, we make, we proposed, etc.

Response 4: Dear reviewer, thank you for carefully reading our manuscript and giving many constructive comments. All words "we" are replaced by "authors" in the revision.

Comment 5. There are many grammatical mistakes and typos in the paper. I have noticed more than 10 mistakes just in the abstract. The authors should revise the overall manuscript carefully to avoid such mistakes.

Response 5: Authors checked and revised the manuscript carefully. Regarding the language of the paper, we will send our manuscript to the English native checker before submitting the final version for publishing.

Comment 6. The technical depth of the paper is not good.

Response 6: Manuscript's organization contains several sections: After Introduction, Section Related studies provide recent works relating to the semantic segmentation task. Section Binary semantic segmentation FCN-VGG-16 introduces the overall architecture of our approach. Finally, Experimental results demonstrate the obtained results and compare our approach with previous methods in the semantic segmentation field. Furthermore, some practical mobile robot strategy application is proven in reference [48].

Comment 7. The current work seems very limited. The authors should do more experiments to present proper outcomes.

Response 7: Authors introduced the proposed method's feasibility in the Experimental results section and current work relating to semantic segmentation in the Introduction. Hence, the proposed method's results are an image containing pixel-wise predictions which can be used for different navigation strategies. Moreover, comparing different FCN architectures using VGG-16 in terms of computational cost and accuracy when performing inference is introduced. Our approach especially yields better results than compared works' methods in the binary semantic segmentation in optimal obstacle avoidance navigation of autonomous mobile robots.

Comment 8. Figure 4 is unable to be read—similarly, check other figures.

Response 8: We checked and revised carefully; please the figure 4 in the manuscript.

Comment 9. In Table 3, what are the 344, 345, 346, and 347? What is the function of these number?

Response 9: The author checked and proved that the problem of "344, 345, 346, and 347" is the word line. Hence, it is the error when the MDPI system transfers the manuscript from the original word file to a pdf file. There is no typo in the original word file. (please see the image as below).

Comment 10. Similarly, in figure 6, what is the function of the number 311?

Response 10: The author checked and proved that the problem of "311" is the word line. Hence, it is the error when the MDPI system transfers the manuscript from the original word file to a pdf file. There is no typo in the original word file. (Please, see the image as below).

Comment 11. Always add a full stop (.) at the end of every caption.

Response 11: We checked and revised carefully.

Comment 12. Please indicate the figures, such as Figure 7a should be Figure 7(a), (b), and so on. Keep consistency in the paper. Revise all the figures.

Response 12: We checked and revised carefully.

Comment 13. Figure 7 results are unable to read like other figures.

Response 13: Dear reviewer, all image results are original. Using a low resource system and using a monocular camera has still ensured the performance of calculation cost and mobile robot navigation strategy based on semantic segmentation.

Comment 14. In a research article, if all the results figures are unable to be read and understood, then how a reviewer can judge the quality of that article?

Response 14: Dear reviewer, all image results are original. The problems are answered and explained in comments 3 and 13.

Comment 15. What is the purpose of Figure 8? It seems useless. Similarly, figure 10.

Response 15: In figure 8, in the front view of the floor plane, the checkerboard image is used in the plane to approximate the transformation matrix. We must measure the correct distance to the obstacle in the practical environment. Hence, the mobile robot navigation will be successfully designed. Moreover, we will perform the approximation based on correspondence points in arbitrary and frontal views of the same checkerboard in figure 10. So, the distance the obstacle in the obtained image will be transferred to the correct length in reality. Finally, the optimal mobile robot path planning will be designed.

Comment 16. What is the function of the dot (.) at the end of equation (4).

Response 16: It means the end of equation (4); we checked and revised it . (Please, see the image as below).

Comment 17. In this work, there are many figures without giving any information's. Those figures must be avoided. They just create confusion for the readers, else nothing.

Response 17: Dear reviewer, we checked and revised carefully.

Comment 18. Figure 15 is completely invisible. Can't be read. Similarly, figures 16 and 17.

Response 18: Dear reviewer, figure 15 illustrates the processing time of each step in optimal mobile robot path planning. So, it strengthened the argument of computational cost in low resource systems. Then figure 16 shows the steering angle when robot tracking path planning. Finally, figure 17 presents the general information of all result steps in the mobile robot navigation strategy.

Round 2

Reviewer 1 Report

Dear authors,

I'm not satisfied with your inclusion of answers to paper's text. You constantly repeat "The main contribution of our paper is to design multi-scale fully convolutional network-based semantic segmentation for mobile robot navigation using low resource system. " which I knew before. 

- You haven't included answer to Comment 4 into the text. Such clear statement of your paper's contributions is what is actually missing in the Introduction.

- The paper should be better organized. 

- You should do more experiments to present proper outcomes or provide significance test (ANOVA, p, t) to prove significance of the results.

- Figure 8 is not well representative. You should add some more examples and then comment it.

- All figures should be explained.

- Section 2 should be reduced.

- You need to calculate mAP in order to see if your ANN is correctly trained. I had some similar problem a month ago and I needed to add more images to training dataset. 

Author Response

RESPONSE TO REVIEWER 1 COMMENTS

Comment 1. You haven't included answer to Comment 4 into the text. Such clear statement of your paper's contributions is what is actually missing in the Introduction

Response 1: Dear reviewer,  the paper tittles “Obstacles Avoidance for Mobile Robot Using Type-2 Fuzzy Logic Controller” [3] is used for the methods of mobile robots navigation at line 42, in Introduction. Then, authors introduce about the solution of complex problems in real-world environments updated and integrated with sensor systems at the line 141-142, in Introduction.

Comment 2. The paper should be better organized. 

Response 2: Dear reviewer, authors revised and organized the manuscript carefully. Regarding the language of the paper, the manuscript was rewritten by English Academic Center. Please, see the revision.

Comment 3. You should do more experiments to present proper outcomes or provide significance test (ANOVA, p, t) to prove significance of the results. 

Response 3: In the Quantitative results, authors first compare our semantic segmentation results to those of prior researchers in Table 1, at the line 575. Then, when moving into the interference area of obstacles, the frontal view may be restricted if the indoor environment contains a large number of obstacles. Consequently, the performance and speed rate of the binary semantic segmentation model satisfy the requirements of [19], Yang et al. [38], and our obtained result using VGG-16 in Table 2, at the line 586. Next, Table 3 presents the quantitative outcomes of our network in response to variations in the encoder block, at the line 597.

Finally, in the practical results, authors use the obtained results based on the fully trained model of binary semantic segmentation FCN-VGG-16, the authors design the following navigation and obstacle avoidance strategy (see Figure 7). After independently examining each component, the authors combine the cost function and produce the final results. The mobile robot avoids the obstacles safely and efficiently in order to reach its destination. Authors evaluate the performance and quality of trajectories by measuring the processing time in Figure 15. The experimental simulation included the following two scenarios: Scenario 1 has one obstacle in the environment's center, while Scenario 2 has two obstacles in the environment's upper right corner. Using our optimal obstacle avoidance strategy described in [48], the processing time in scenario 1 is 0.0493 seconds and scenario 2 is 0.0725 seconds. Furthermore, our obstacle avoidance strategy [45] ensured that the steering angle changes were less than 0.2 rad, in Figure 16.

Authors successfully apply perspective correction to the segmented image in order to construct the frontal view of the general area, which identifies the available moving area. The optimal obstacle avoidance strategy is comprised primarily of collision-free path planning, reasonable processing time, and smooth steering with low steering angle changes.

Comment 4. Figure 8 is not well representative. You should add some more examples and then comment it.

Response 4: Dear reviewer, authors revised it carefully. After performing binary semantic segmentation, the authors obtained the images in Figure 8 labeled with the available area for movement (white) and unavailable space (black). The perspective of captured images produces significant distortion. So measuring the distance to the obstruction was quite difficult. In the front view of the floor plane, the image of a checkerboard is utilized to approximate the transformation matrix.

Authors have to obtain perspective correction to the segmented image in order to construct the frontal view of the general area, which identifies the available moving area. Then, the optimal obstacle avoidance strategy will be constructed successfully.

Comment 5. All figures should be explained.

Response 5: Dear reviewer, authors revised and explained about all figures carefully.

Comment 6. Section 2 should be reduced.

Response 6: Dear reviewer, authors revised and rewrote Section 2 carefully.

Comment 7. You need to calculate mAP in order to see if your ANN is correctly trained. I had some similar problem a month ago and I needed to add more images to training dataset. 

Response 7: Dear reviewer, the main contribution of our paper the design of multi-scale fully convolutional network-based semantic segmentation for mobile robot navigation using low resource system. So, authors collected additional real images to enhance the dataset. As for proof of feasibility of our proposed method,  authors conduct the final network performance test on an image containing numerous objects in the background. Our network accurately predicts the ground boundary in these difficult circumstances, demonstrating its robustness across corridor types in 4.1 (Qualitative results). Moreover, using only the single camera system for mobile robot navigation, authors used mIoU for the semantic segmentation task and comparison with prior methods in 4.2 (Quantitative results) and 4.3 (Practical results).
Please see the attachment.

Reviewer 2 Report

No more

Author Response

We sincerely thank the reviewers for their positive feedback and constructive comments. The reviewers' suggestions and comments are beneficial to guide us in improving our manuscript.

Round 3

Reviewer 1 Report

You could add clear version of the manuscript as well. It would make easier to understand the final outline of the paper. 

Some of your comments could be useful to readers and could be included in final version.